# A Study on Farmers’ Participation in Environmental Protection in the Context of Rural Revitalization: The Moderating Role of Policy Environment

**DOI:** 10.3390/ijerph20031768

**Published:** 2023-01-18

**Authors:** Hao Dong, Yang Zhang, Tianqing Chen

**Affiliations:** 1Institute of Land Engineering and Technology, Shaanxi Provincial Land Engineering Construction Group Co., Xi’an 710075, China; 2School of Management, Xi’an Jiaotong University, Xi’an 710049, China

**Keywords:** rural environmental protection, farmers’ participation behavior, farmers’ participation intention, extended theory of planned behavior, PLS-SEM

## Abstract

This study investigates the environmental protection behavior of farmers in the Guanzhong Plain region and the factors influencing their participation, in order to improve the enthusiasm of farmers’ participation and promote the formation of “good governance” in rural ecological environments. Based on interviews with 295 farmers, the influence of psychological cognitive factors on farmers’ intention and behavior to participate in environmental protection was analyzed using partial least squares structural equation modeling under the extended theory of planned behavior, and the moderating effect of policy environment in the relationship between farmers’ intention to participate and participation behavior was revealed. The research results show that: (1) The current situation of farmers’ participation in environmental protection is generally characterized by “strong intentions and weak actions”. (2) Participation consciousness and benefit perception have a greater impact on farmers’ intention to participate in environmental protection, perceived behavioral control has a smaller impact, and subjective norms do not have a significant impact. (3) Perceived behavioral control and participation intention have a greater influence on farmers’ participation in environmental protection behavior, subjective norms have less influence, and there is no direct influence of perceived benefits and participation awareness on farmers’ participation behavior, i.e., farmers’ participation intention has indirect influence on participation behavior. (4) The moderating effect of policy environment indicates that policy environment has a significant positive effect on the relationship between farmers’ participation intention and participation behavior.

## 1. Introduction

The rural revitalization strategy sets out the goal of being ecological livable, which points out the direction for the development of rural environmental governance in China. Therefore, at present, it is urgent to explore a new way to promote rural environmental protection to achieve good governance, and “participation” is the main way to form a pattern of “good governance” of rural ecological environments. The natural “presence” of farmers determines that the effective governance of rural ecological environments cannot be separated from the participation of farmers [1]. Fei Meng et al. [2] believe that “without the participation and contribution of farmers, environmental protection cannot be realized”. However, at present, farmers’ participation in environmental protection is characterized by high attention and low participation [3]. In order to ensure the orderly participation of farmers, the No. 1 Central Document issued by the Chinese government for 2020 and 2022, respectively, mentioned mobilizing farmers to participate in the control of rural ecological and environmental pollution, and extensively mobilizing farmers to participate in rural revitalization.

How to stimulate the endogenous power of farmers’ participation is a problem worth thinking about in rural environmental protection. Existing studies have pointed out that interpersonal trust, institutional trust [4], cadre relationship [5], and resource endowment characteristics of farmers [6] all have an impact on farmers’ intention to participate in environmental protection. However, according to behavioral economics theory, cognition is a prerequisite for individual behavior. In recent years, scholars have conducted extensive research on farmers’ cognition and behavioral response to rural environmental protection based on the theory of planned behavior. In terms of farmers’ cognition, Wang et al. [7] showed that subjective norms significantly affected farmers’ intention to participate in non-point source pollution control, while intention and perceived behavior control had key influences on their participation. Lu et al. [8] added two factors of perceived benefit and perceived cost on the basis of the theory of planned behavior to analyze the impact of farmers’ psychological cognition on their green production behavior. Dong et al. [9] divided farmers’ green production behavior into incentive behavior and constraint behavior, and analyzed the influence of farmers’ cognition, behavioral attitude, subjective norm, perceived behavioral control, and production intention on green agricultural development policy on their behavior. In terms of farmer behavior, it mainly emphasizes the participation of farmers in the whole process of rural environmental protection. According to Lin et al. [10], farmer participation is an active process in which farmers influence the implementation and direction of rural development projects, which mainly includes farmer involvement in decision making, decision-making implementation, project supervision, and evaluation of development projects. Accordingly, the behavior of farmers participating in rural environmental protection can be divided into decision-making behavior, protection behavior, and supervision behavior. The rural revitalization strategy has been implemented for nearly five years, and the policy has significantly affected farmers’ intention and behavior. In different policy environments, the effect of participation intention in environmental protection on participation behavior may be different [11].

To sum up, there are abundant research results on farmers’ participation in rural environmental protection, but there is still a gap for further expansion. First, few scholars have subdivided and investigated the decision-making, protection, and supervision behaviors of farmers’ participation in environmental protection. Second, there are few studies on the moderating effects of guiding policies, incentive policies, and restraint policies in the policy environment on participation intention and participation behavior. In view of this, this study divides farmers’ participation in environmental protection behavior into decision-making behavior, protection behavior, and supervision behavior, and introduces benefit perception and participation consciousness into the theory of planned behavior. Based on the survey data of 295 farmers in Guanzhong Plain, a partial least square structural equation model was established to explore the effects of psychological cognitive factors on farmers’ intention and behavior to participate in environmental protection, as well as the moderating effects of policy environment on their intention and behavior to participate in environmental protection. The Baoji Fengxiang and Weinan Linwei districts in Guanzhong Plain were selected as the research area in this study. As these areas are major grain-producing areas in five provinces in western China, with the influence of human factors in the construction of high-standard farmland projects, environmental pollution poses a serious threat to the rural ecological environment, and farmers are important actors in rural environmental protection. Therefore, it is of great significance to study the influencing factors of farmers’ participation in environmental protection to promote farmers’ active participation and protect the environment in major grain-producing areas.

The structure of the article is as follows: Section 2 summarizes the theory and hypotheses; Section 3 introduces the questionnaire and data source; Section 4 presents the results of the study; and Section 5 summarizes the conclusions and contributions, and provides some practical implications of the empirical findings.

## 2. Theory and Hypotheses

### 2.1. Theory of Planned Behavior (TPB)

TPB is a theory by Ajzen [12] based on rational behavior theory and multi-attribute attitude theory to add human perceptual control over behavior results, which provides an influence path of “cognition–intention–behavior”, holding that attitude (ATT), subjective norm (SN), and perceived behavioral control (PBC) are the three main factors that determine behavioral intention (BI), while behavioral response is directly affected by intention and behavioral control [13]. In rural environmental protection, farmers’ participation behaviors are complicated, and their behaviors are affected by psychological cognition, resource endowment, risk preference, objective environment, and other factors [14,15,16]. Therefore, understanding the psychological cognition of farmers and the policy environment (PE) of their participation is the first step to explain and predict their participation behavior. [17]. Among them, ATT can be used to explain the effect of farmers’ participation intention by benefit perception (BP). Participation consciousness (PC), as an important part of psychological cognitive factors, can stimulate farmers’ participation motivation. SN and PBC reflect the influence of social influence and control ability on farmers’ participation intention (PI) and participation behavior (PB). PI as an intermediary variable explains the influence of farmers’ psychological cognition on PB. PE reflects the difference of PI on PB under different policy backgrounds. Therefore, based on the extended planning behavior theory, the research hypothesis is proposed and the theoretical model is constructed (Figure 1).

### 2.2. Benefit Perception (BP)

ATT is an individual’s expectation of the positive or negative attributes of a behavior result [18]. In rural environmental protection, ATT of farmers can be divided into economic rationality and ecological rationality [19], which are explained by the BP of economic benefits and ecological benefits [20]. In terms of economic rationality, farmers’ participation in environmental protection can improve the quality of the ecological environment, which helps to promote the development of rural tourism and ecological agriculture, and increase the economic income of farmers. Therefore, when farmers believe that environmental protection can increase their economic benefits, they are more willing to participate in environmental activities. In terms of ecological rationality, farmers’ participation in environmental protection can improve the environmental quality of water, soil and air, as well as reduce soil and water loss and prevent land desertification in farming areas. The higher the farmers’ perception of ecological benefits generated by environmental protection, the higher their PI. Accordingly, Hypothesis 1 is proposed:

**Hypothesis 1 (H1).** 
*BP has a positive effect on PI in environmental protection.*


### 2.3. Participation Consciousness (PC)

Farmers’ awareness of environmental protection participation can be measured by PC [21]. As the main body of agricultural production and rural life, farmers are direct participants and beneficiaries. Only when farmers realize that rural environmental protection cannot be separated from their participation can they change their indifferent attitude and gain a stronger desire to participate [22]. In addition, farmers have a weak awareness of their own rights, and do not fully realize that a benefit of participating in rural decision making and protecting the ecological environment from destruction is to safeguard their own rights and interests, which is also an important factor affecting the low participation enthusiasm of most farmers [23]. Accordingly, Hypothesis 2 is proposed:

**Hypothesis 2 (H2).** 
*PC has a positive effect on PI in environmental protection.*


### 2.4. Subjective Norm (SN)

SN reflects the importance attached to the influence of others or groups on individual behavioral decision making [24]. In rural environmental protection, SN mainly come from the “herd effect” of media publicity, village cadres’ call, and villagers’ neighborhood demonstration [25]. If those who are important to farmers participate in environmental protection, or think it is good to participate in environmental protection, farmers will feel the pressure from the outside and have a higher PI and PB. Accordingly, Hypotheses 3 and 4 are proposed:

**Hypothesis 3 (H3).** 
*SN has a positive effect on PI in environmental protection.*


**Hypothesis 4 (H4).** 
*SN has a positive effect on PB in environmental protection.*


### 2.5. Perceived Behavior Control (PBC)

PBC refers to an individual’s perceived control ability when performing a certain behavior [26]. As the affairs of environmental protection are complex and highly professional, farmers’ participation in the decision-making, implementation, and supervision processes requires high cultural quality, knowledge, and skills [27]. Participation opportunity is a prerequisite for farmers’ participation. The expression of farmers’ views and opinions in environmental protection requires the local government and village cadres to solicit opinions from villagers extensively [28]. Participation channels are essential for farmers to participate, and diversified, unimpeded, and convenient channels can promote the enthusiasm of farmers to participate [29]. Accordingly, Hypotheses 5 and 6 are proposed:

**Hypothesis 5 (H5).** 
*PBC has a positive effect on PI in environmental protection.*


**Hypothesis 6 (H6).** 
*PBC has a positive effect on PB in environmental protection.*


### 2.6. Participation Intention (PI)

PI refers to the motivation intensity of an individual to adopt a certain behavior [30]. Studies have shown that there is a highly positive correlation between individual behavioral intention and behavioral response; that is, the stronger the individual behavioral intention, the more likely the individual is to perform a certain behavior [31]. In rural environmental protection, PI reflects the intensity of their motivation to participate in environmental protection [32]. The stronger their PI in environmental protection, the greater the probability of their participation in environmental protection. Accordingly, Hypothesis 7 is proposed:

**Hypothesis 7 (H7).** 
*PI has a positive effect on PB in environmental protection.*


### 2.7. Policy Environment (PE)

In different policy environments, the effect of PI in environmental protection on PB may be different. First of all, in terms of guiding policies, the local government should publicize relevant policies to farmers, as well as publicizing ecological and environmental information, so that farmers can be correctly guided, fully grasp the environmental protection information, and supervise the governance process [33]. Secondly, economic punishment or criticism education in village rules and regulations can restrain farmers’ behaviors that damage the ecological environment, and economic rewards or honors can encourage farmers to participate in environmental protection. Finally, in terms of incentive policies, the more responsive the government is, the more farmers feel valued and respected, and the more willing they are to take practical actions [34]. In addition, government subsidies can help farmers reduce the burden of environmental protection capital investment, reduce their participation costs, and encourage farmers to actively participate in environmental protection. Accordingly, Hypothesis 8 is proposed:

**Hypothesis 8 (H8).** 
*PE plays a positive moderating role between PI and PB.*


## 3. Methodology

### 3.1. Questionnaire Design

Based on the theory of planned behavior, this paper sets up seven latent variables: BP, PC, SN, PBC, PW, PE, and PB. The observed variables of each latent variable were measured using a 7-point Likert scale (1 = strongly agree, 7 = strongly disagree) (Appendix A Table A1).

### 3.2. Data Collection

This study is based on the data of farmers in Baoji and Weinan, Guanzhong region, from April to September 2022. The reason this research area was selected is that high-standard farmland projects are being built there, and the research area has accumulated rich experience in environmental protection. The villagers of the villages participating in environmental protection in the above areas were selected as the investigation objects. The survey was conducted from April to June and from July to September 2022. First, 30 farmers were selected from the field of high-standard farmland project for pre-survey, and then the validity of the questionnaire was tested and the snowball method was adopted to collect the questionnaire. The questionnaire was distributed by a combination of interviews and on-site questionnaires. The interviewees were villagers or village officials involved in or familiar with environmental protection. In order to improve the authenticity of respondents filling in the questionnaire, and to clearly express the academic purpose and research value of the research to respondents during the research process, we allowed respondents to fill in and handle the questionnaire anonymously, so as to ensure that the responses were voluntary and free from relevant concerns, as well as ensure the reliability and persuasion of the results. A total of 400 questionnaires were distributed in this survey, with 295 valid samples (effective rate of 73.75%). 

### 3.3. Data Analysis

PLS-SEM was conducted by using the SmartPLS 3.0 Software Tool (Version 3.3.6) to analyze and interpret the model. The partial least square structural equation model (PLS-SEM) is mainly used to model the relationship between latent variables [35], which can effectively solve the problems that are difficult to directly observe, such as farmers’ cognition, clearly describe the decision-making process of farmers’ behavior, and analyze the influence of variables and whether there are differences [36]. We divided the analysis into two parts: firstly, the latent construct dimensions, validity and reliability of the measurement model were evaluated; secondly, the path coefficient and path significance of the structural model were evaluated. The PLS algorithm was used to derive the path coefficients of the structural model, and the path weighting scheme algorithm was used to give the standardized regression coefficients [37]. The statistical significance of the structural path was evaluated by Bootstrapping procedure.

### 3.4. Sample Characteristics

As shown in Table 1, 295 valid questionnaires were collected. The characteristics of the samples were as follows: 182 (61.7%) males and 195 (66.1%) middle-aged adults over 50 years old. Of these, 215 (72.9 percent) had less than a high school education, accounting for nearly three-quarters of the sample size. Only 26 of them were village cadres (the administrator or decision maker of the village). The results of mean and variance of latent variables are shown in Table 2: BP (M = 5.000, SD = 1.082), PC (M = 5.212, SD = 1.052), SN (M = 4.714, SD = 1.116), PBC (M = 4.894, SD = 1.148), PI (M = 4.898, SD = 1.112), PE (M = 5.223, SD = 1.176), and PB (M = 5.066, SD = 1.082).

### 3.5. Common Method Variance

According to Podsakoff’s method, process control and statistical control were adopted in this study to reduce homologous bias. In the process control stage, it was not clearly marked that all the study variables belong to independent variables, and the variables were also mediating and regulating variables. Therefore, subjects did not deliberately violate their own subjective judgment in order to suit the research topic. In addition, using the Harman test to test for the threat of common method bias, the first principal component explained only 25.3% (<50% threshold value) of the total variance according to exploratory factor analysis. Due to this, the first factor does not solve for the large number difference, so the threat of common method deviation does not exist. Then, the method factor was introduced to test and a two-factor model was established; that is, a common method factor was added to the structural equation model as a global variable, and the fitting degree changes of the skew factor model and the two-factor model were compared. The results showed that, after controlling the common method factors, the model fit degree did not change significantly, so the homologous variance problem was not serious.

### 3.6. Reliability and Validity

For reliability and validity tests, Cronbach’s α coefficient was used to test the internal consistency of each item, and a reliability coefficient was constructed to test the internal quality of potential variables. As shown in Table 3, Cronbach’s α coefficients of the four latent variables were all greater than 0.7, indicating good reliability of the scale. Scale validity mainly includes content validity and structure validity. The scale design of this study was based on previous studies by other scholars, so the content validity is good. The discriminative validity includes convergence validity and structure validity. First, convergence validity was measured using mean variance extraction (AVE) and structural validity was measured using reliability analysis (CR). As shown in Table 4, AVE of latent variable is greater than 0.50 [37], meeting the requirement of critical value 0.5, and CR value is greater than 0.80 [35], indicating that the variable has good convergence validity and structural validity. Secondly, as shown in Table 4, the square root of variance of each variable is greater than the correlation coefficient between the row and the column, indicating that this measure has good discriminant effectiveness.

## 4. Results

### 4.1. Path Analysis

This paper used SmartPLS 3.0 software (Version 3.3.6) to construct a partial least square structural equation model of farmers’ participation in environmental protection behavior, as shown in Figure 2 and Table 5. The results of path analysis showed that BP, PC, SN, and PBC had positive effects on PI (β = 0.267, *p* < 0.001; β = 0.438, *p* < 0.001; β = 0.075, *p* > 0.05; β = 0.125, *p* < 0.05); H1, H2, and H5 were all verified. H3 was not verified. The results of path analysis showed that SN, PI, and PBC had positive effects on PB (β = 0.166, *p* < 0.01; β = 0.525, *p* < 0.001; β = 0.166, *p* < 0.01); H4, H6, and H7 were all verified. This may be because most of the farmers think that whether they participate in rural environmental protection depends on the publicity of the media, the call of the village leaders, and the demonstration of the villagers’ neighbors. If these important organizations or others participate, the farmers will also participate, which is the so-called “herding effect” psychology, such that the influence of important organizations or others directly affects farmers’ participation behavior. The R^2^ of PI suggests that BP, PC, SN, and PBC explain the 45.7% effect of PI. The R^2^ of PB suggests that PBC, SN, and PI explain the 54.8% effect of PB. It can be seen that the explanatory power of this model is strong. By observing Q^2^ of endogenous variables to evaluate the predictive ability of the model, the results show that PI (Q^2^ = 0.362) and PB (Q^2^ = 0.370) are all greater than 0, and the model has good predictive correlation.

### 4.2. Moderating Effect

It was assumed that the PE has a moderating effect between farmers’ PI and PB. Therefore, the policy of guidance, incentive, and constraint was introduced to investigate the regulating effect of PE on farmers’ PI and PB. Firstly, the guiding policy was measured by two indicators: environmental publicity and education (PE1), and ecological and environmental information disclosure (PE2). The incentive policy was measured by two indicators: the degree of local government response (PE3), and government subsidy policy (PE4). The incentive and restraint policies were measured by an index called village Rules and Covenants (PE5). Referring to the study by Wen Zhonglin et al., using the topic packaging strategy, PE6 = (PE1 + PE2)/2 and PE7 = (PE3 + PE4)/2 were used to integrate the two indicators in the guidance policy and incentive policy, respectively, to form the mean values of PE6 and PE7, which were taken as new observation variables. At the same time, PE5, which has incentive and restraint policies, was used as an indicator of PE. At this time, the independent variable PI and the regulating variable PE have three indicators, respectively: PI1, PI2, and PI3, and PE5, PE6, and PE7. According to the product index pairing strategy, confirmatory factor analysis was performed for PI’s and PE’s three respective indexes. Then, the standardized factor loads were multiplied from high to low pairing according to the strategy of “large pairing large, small pairing small”, and a set of pairing product indexes of PI2PE7, PI3PE6, and PI1PE5 were obtained. Using this pairing strategy in the research, the interaction product term generated had the highest reliability. By using this strategy, the interaction term between PI and PE was obtained, and the moderating effect of PE was tested. The results are shown in Table 5. PI, PE, and the interaction terms between PI and PE have a significant positive influence on farmers’ PB at the level of 0.1%, indicating that PE has a significant strengthening effect on the relationship between PI and PB, and a positive moderating effect of PE exists. It shows that the better the PE, the more likely it is farmers’ PI in environmental protection will be transformed into PB. Therefore, H8 is verified.

## 5. Conclusions

In this study, based on the survey data of farmers in Guanzhong Plain under the framework of extended planned behavior theory, the partial least square structural equation model was used to explore the mechanism of the influence of psychological cognitive factors on farmers’ PI and PB in environmental protection. It also revealed the moderating effect of PE on farmers’ PI–PB. The conclusion is as follows:(1)Farmers’ PI in environmental protection is very high, but their intention to participate in decision making and supervision of environmental protection is not high. However, farmers’ participation in decision-making behavior, protection behavior, and supervision behavior is low, showing the characteristics of “strong will and weak action”.(2)BP, PC, and PBC have significant positive effects on PI in environmental protection, but SN has no significant effects. The order of effect was PC (0.438) > BP (0.267) > PBC (0.125). That is, the higher the farmers’ PC, BP, and PBC, the stronger the farmers’ PI in environmental protection.(3)SN, PBC, and PI have significant positive effects on farmers’ PB. The order of effect was PI (0.525) > SN (0.209) > PBC (0.199). The effect of BP and PC on farmers’ PB was realized through PI transmission. That is, the greater the PBC, the higher the PI, and the greater the impact of SN, the stronger the farmers’ PB.(4)PE has a significant positive moderating effect on the relationship between farmers’ PI and PB. That is, the higher the degree of government response, the greater the degree of government subsidy, the greater the degree of environmental protection publicity and education, the greater the degree of ecological environment information disclosure, the stronger the incentive and restraint of village rules and regulations, the stronger the positive effect of farmers’ participation intention on their participation behavior.

## 6. Discussions

### 6.1. Theoretical Implications

This study is able to theorize the SN, BP, PC, and PBC of farmers, thus unlocking the development path of environmental protection behavior. This study also theorizes that farmers’ perception of the policy environment will affect their participation behavior in the context of rural revitalization. For this purpose, we adopt the hybrid model, namely the extended TPB model. This extended model has been successfully used to explore farmer willingness and farmer behavior [9]. The rural revitalization strategy has put Chinese farmers and rural China on the “front line” of environmental protection and governance [29]. This study provides a new contribution in this context. This study examines how BA creates barriers to farmers’ participation in environmental behavior, and explains how the extended TPB model will help explain this situation.

### 6.2. Practical Implications

Protect farmers’ environmental rights and promote their enthusiasm to participate in decision making and supervision. When making decisions on environmental protection in rural areas, the government needs to ensure that rural households can participate in the consultation on, suggestion, and decision making of environmental protection issues in the form of discussion by village committees through villagers’ meetings or village people’s congresses. It is also necessary to foster environmental NGOs to represent, organize, and drive farmers to fully participate in supervision, so as to make up for the lack of farmers’ participation and safeguard their environmental rights and interests. Strengthen environmental protection publicity and education, and enhance farmers’ awareness of environmental protection. The government should actively promote environmental protection knowledge and the idea that “Lucian waters and lush mountains are invaluable assets” by hosting public environmental protection activities and lectures on ecological and environmental protection knowledge. In addition, the government should carry out regular training on agricultural skills, guide farmers to rationally apply chemical fertilizers and pesticides, and actively participate in garbage sorting and recycling, so that farmers can enhance their environmental awareness through learning and practice. Set an example and lead farmers to participate in environmental protection. The government should give full play to the exemplary and leading role of village cadres in environmental protection, encourage and guide all villagers to participate in environmental protection, and give public praise or material rewards to farmers with excellent performance in rural environmental protection, to mobilize the enthusiasm of farmers to participate. Provide participation opportunities and broaden participation channels to improve farmers’ behavior control ability. The government should provide diverse and convenient channels for participation, such as environmental protection hotlines, mailboxes for people’s suggestions, environmental questionnaires, etc., to collect opinions and suggestions from rural households, so that rural households have opportunities and channels to participate in rural environmental protection decisions and supervision actions. Create a good policy environment to improve the behavioral response degree of farmers. The government should strengthen the disclosure of regional and practical environmental information on the status of rural ecological environments and soil quality through broadcasting, announcements, meetings, and other forms. It is also necessary to attach importance to the regulating role of the incentive and constraint mechanism of village rules and regulations on farmers’ behavior, and organize farmers to participate in the formulation of village rules and regulations in line with the actual situation of the village.

### 6.3. Limitations and Future Research

Although snowball sampling was used to conduct the sample survey, the samples were mainly limited to Baoji and Weinan, Shaanxi Province, which, to some extent, affected the representativeness of the samples and the accuracy and generalization of the research findings. In the future, more samples can be selected nationwide for empirical testing, so as to revise, expand, and improve the measurement scale and theoretical model proposed in this paper. In addition, the empirical research carried out in this paper adopts cross-sectional data, while the questionnaire survey collects data at roughly the same time point, which does not involve dynamic simulation of the effects of different policies. Therefore, in the future, a longitudinal research method can be adopted to collect time series data, further introduce system dynamics, dynamic evolutionary game, and other methods to carry out dynamic policy effect evaluation, explore the effects of different policies and their combinations on farmers’ environmental protection behaviors, and provide a decision-making basis for further village ecological governance.

## Figures and Tables

**Figure 1 ijerph-20-01768-f001:**
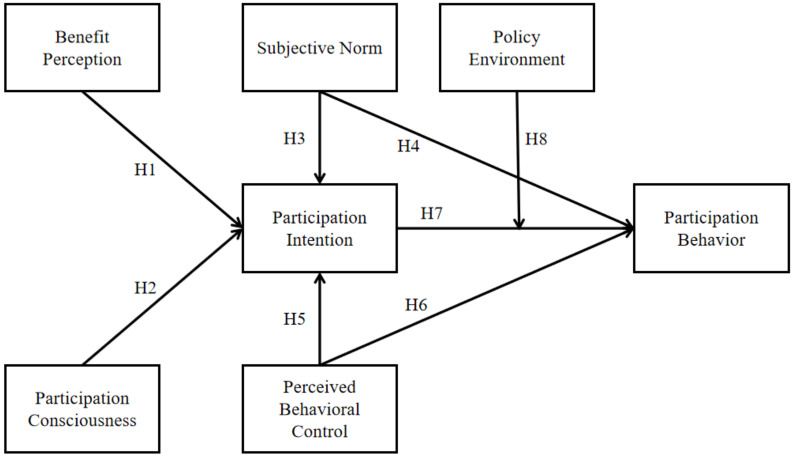
Research model.

**Figure 2 ijerph-20-01768-f002:**
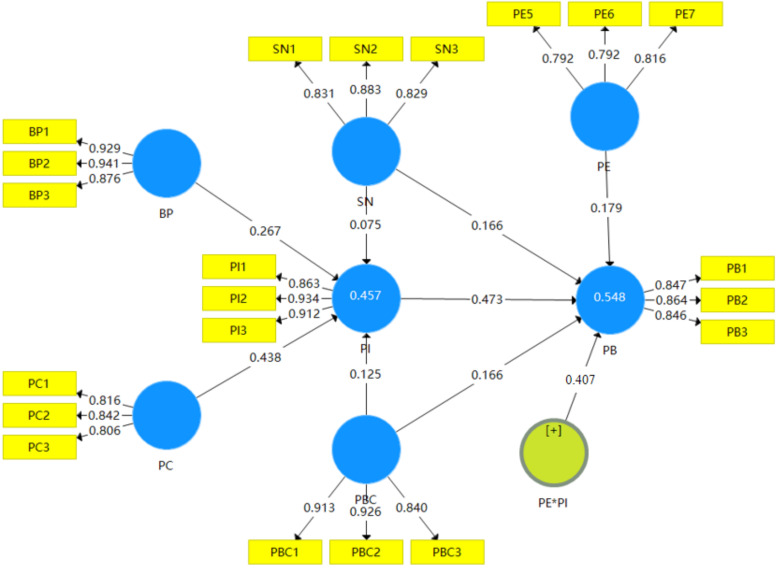
Path diagram of modified model.

**Table 1 ijerph-20-01768-t001:** Sample characteristics.

Variables	Definitions	Frequency	Proportion
Gender	Female	113	38.3%
	Male	182	61.7%
Age	20–30 years old	15	5.1%
	31–40 years old	24	8.1%
	41–50 years old	61	20.7%
	51–60 years old	185	62.7%
	61 years old and above	10	3.4%
Education	Not been to school	45	15.3%
	Primary school	122	41.4%
	Middle school	58	19.7%
	high school	70	23.7%
	College	10	3.4%
Cadre	Cadre	26	8.8%
	Non-cadre	274	91.2%

**Table 2 ijerph-20-01768-t002:** Descriptive statistics.

Variables	No. Items	Mean	SD
Benefit Perception	3	5.000	1.082
Participation Consciousness	3	5.212	1.052
Participation Intention	3	4.898	1.112
Subjective Norm	3	4.714	1.116
Perceived Behavior Control	3	4.894	1.148
Policy Environment	5	5.223	1.176
Participation Behavior	3	5.066	1.082

Note: SD represents the standard deviation.

**Table 3 ijerph-20-01768-t003:** Reliability and validity.

Variables	Items	Loadings	Cronbach’s α	CR	AVE	VIF
Benefit Perception	BP1	0.929	0.905	0.940	0.839	3.184
	BP2	0.941				3.740
	BP3	0.876				2.470
Participation Behavior	PB1	0.848	0.812	0.888	0.726	1.866
	PB2	0.865				1.967
	PB3	0.844				1.623
Perceived Behavior Control	PBC1	0.831	0.805	0.885	0.726	1.705
	PBC2	0.883				1.987
	PBC3	0.829				1.657
Participation Consciousness	PC1	0.816	0.759	0.861	0.674	1.524
	PC2	0.842				1.561
	PC3	0.806				1.512
Policy Environment	PE1	0.766	0.802	0.864	0.559	1.665
	PE2	0.783				1.696
	PE3	0.750				1.591
	PE4	0.734				1.599
	PE5	0.703				1.568
Participation Intention	PI1	0.863	0.888	0.930	0.817	2.123
	PI2	0.934				3.628
	PI3	0.912				2.980
Subjective Norm	SN1	0.913	0.874	0.923	0.799	2.731
	SN2	0.926				3.033
	SN3	0.840				1.948

Note: CR: Composite Reliability; AVE: Average Variance Extracted; VIF: Variance Inflation Factors.

**Table 4 ijerph-20-01768-t004:** Discriminant validity—Fornell-Larcker Criterion and Heterotrait-Monotrait Ratio.

Variables	BP	PB	PBC	PC	PE	PI	SN
Benefit Perception (BP)	**0.916**	0.467	0.163	0.389	0.374	0.512	0.410
Participation Behavior (PB)	0.399 **	**0.852**	0.511	0.819	0.680	0.779	0.524
Perceived Behavior Control (PBC)	0.140 **	0.414 **	**0.848**	0.514	0.491	0.369	0.240
Participation Consciousness (PC)	0.322 **	0.642 **	0.400 **	**0.821**	0.696	0.724	0.415
Policy Environment (PE)	0.319 **	0.550 **	0.391 **	0.540 **	**0.748**	0.605	0.380
Participation Intention (PI)	0.464 **	0.667 **	0.313 **	0.597 **	0.510 **	**0.904**	0.440
Subjective Norm (SN)	0.365 **	0.446 **	0.204 **	0.345 **	0.322 **	0.389 **	**0.894**

Note: ** Correlation is significant at the 0.01 level (2-tailed), bold diagonal entries are square root of AVEs, and Heterotrait–Montrait ratios (HTMT) (Underlined) are below 0.85.

**Table 5 ijerph-20-01768-t005:** Results of hypothesis testing.

Hypothesis	Effect	Path	Path Coefficient	STDEV	Lower (2.5%)	Upper (97.5%)	t-Statistics	*p*-Value	Decision
Direct Relationships									
H1	Direct	BP -> PI	0.267	0.057	0.154	0.375	4.668	0.000 ***	Accept
H2	Direct	PC -> PI	0.438	0.076	0.277	0.574	5.773	0.000 ***	Accept
H3	Direct	SN -> PI	0.075	0.05	−0.017	0.179	1.486	0.137	Rejection
H4	Direct	SN -> PB	0.166	0.054	0.072	0.285	3.221	0.004 **	Accept
H5	Direct	PBC -> PI	0.125	0.059	0.017	0.245	2.129	0.033 *	Accept
H6	Direct	PBC -> PB	0.166	0.055	0.055	0.269	3.127	0.004 **	Accept
H7	Direct	PI -> PB	0.525	0.078	0.282	0.584	5.724	0.000 ***	Accept
Mediating Relationships									
H1 * H7	Indirect	BP -> PI -> PB	0.119	0.026	0.065	0.167	4.585	0.000 ***	Accept
H2 * H7	Indirect	PC -> PI -> PB	0.195	0.061	0.085	0.32	3.189	0.001 ***	Accept
H3 * H7	Indirect	SN -> PI -> PB	0.033	0.022	−0.007	0.082	1.477	0.14	Rejection
H5 * H7	Indirect	PBC -> PI -> PB	0.055	0.027	0.007	0.111	2.085	0.037 *	Accept
Moderating Relationship									
H8	Indirect	PE * PI -> PB	0.407	0.051	0.212	0.523	4.892	0.000 ***	Accept
SRMR composite model = 0.061 R^2^_PI_ = 0.457; Q^2^_PI_ = 0.362 R^2^_PB_ = 0.548; Q^2^_PB_ = 0.370									

Note: Significant level: *p* < 0.10; * *p* < 0.05; ** *p* < 0.01; *** *p* < 0.001.

## Data Availability

The data presented in this study are available within the article.

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
