# Peer review of "A Study on Farmers’ Participation in Environmental Protection in the Context of Rural Revitalization: The Moderating Role of Policy Environment"

_ijerph, 2023, doi:10.3390/ijerph20031768_

Round 1
Reviewer 1 Report
Dear Authors,
The manuscript entitled The impact of farmer participation on environmental protection behavior in the era of rural Revitalization: The moderating role of policy environment deals with an interesting and current topic and it gives valuable information on farmers behaviour for environmental protection in a significant rural area from China.
The study is well written, the discussion is balanced, but some important information is missing and it needs some improvements. Please see the attached file.

Author Response
Response to Reviewer 1 Comments
On behalf of my co-authors, we thank you very much for giving us an opportunity to revise our manuscript, we appreciate editor and reviewers very much for their positive and constructive entitled “The impact of farmer participation on environmental protection behavior in the era of rural Revitalization: The moderating role of policy environment”.
We have studied reviewer’s comments carefully and have made revision which marked in red in the paper. We have tried our best to revise our manuscript according to comments. Attached please find the revised version, which we would like to submit for your kind consideration.
We would like to express our great appreciation to you and reviewers for comments on our paper.
Looking forward to hearing from you.
Thank you and best regards.
Point 1: Concepts are not always clearly defined or the use of the English language is difficult to understand. I am not sure that I understand correctly what the authors mean by ‘policy environment’. After reading the paper, I suppose you refer to the policies of environmental protection. If this is the case, I strongly recommend rephrasing and correcting it throughout the paper.
Response 1: In different "policy environments", the effect of farmers' willingness to participate in environmental protection on their participation behavior may be different. The "policy environment" we refer to is that only the village or county government of the farmer provides "guiding policies, incentive policies and restraint policies".
Point 2: Also, farmer participation on environmental protection behaviour is also a bit odd and not very clear.
Response 2: Our title is "The impact of farmer participation on environmental protection behavior in the era of rural Revitalization: "The moderating role of policy environment", we know the influence of intention on behavior in the theory of planned behavior. We conducted a study on farmers' participation in environmental protection behavior under the implementation of "rural revitalization" in specific Chinese national conditions. We should change the title to "A study on farmers' Participation in environmental protection in the context of Rural Revitalization: The moderating role of policy environment"
Point 3: Moreover, in subchapter 2.1, the authors state that (lines 107-109): o ‘Understanding farmers' psychological cognition of environmental protection participation and the policy environment (PE) of participation is the first step to explain and predict farmers' participation behaviors’. I find it difficult to understand your line of though here and I can only hope I can grasp the correct idea.
Response 3: In the governance of rural ecological environment, farmers' participation behavior is relatively complex, and its behavior is affected by psychological cognition, resource endowment, risk preference, objective environment and other factors. Therefore, understanding the psychological cognition of farmers and the policy environment of their participation is the first step to explain and predict their participation behavior. Policy environment is affected by farmers' perceived risk preference and objective environmental factors.
Point 4: Environmental policy in rural China. The title of the paper makes reference to ‘the moderating role of policy environment’, but apart from some information in sub-chapter 2.7 Policy environment, lines 179-192, which is strictly related to the work hypothesis, the authors do not acknowledge this issue in the Chinese context.
Response 4: Dear reviewer, what our team studies in this study is "the moderating effect of policy environment", which is different from "environmental policy". Because in Chinese context, we refer to the "policy environment" which is decided jointly by the government and the village collective rules, and the "environmental policy" is formulated by the government and implemented by farmers.
Point 5: What are the rural environmental problems? Please detail the current environmental policy background and arrangements, focusing particularly on the rural area on the whole and on the study region in particular. What are the specific policy tools for this type of policy?
Response 5: We also indicated in the questionnaire that the question items involved in our policy environment: (1) Environmental publicity and education; (2) Ecological environment information disclosure; (3) The level of local government response; (4) Government subsidy policy; (5) The influence of village rules and conventions. The "policy environment" we study involves government incentive policies, village collective and policy guidance policies, and government and village regulations restraint policies.
Point 6: - Study area – there should be a short presentation of the study area in terms of social and economic background and environmental issues (the major environmental problems are related only to agricultural production? What other sources are there?).
Response 6: Based on the food security strategy, China has implemented the strictest "red line" policy on cultivated land, which means that major grain-producing areas generally do not have pollution other than production pollution. Baoji Fengxiang and Weinan Linwei districts in Guanzhong Plain are selected as the research area in this study. As these areas are major grain-producing areas in five provinces in western China, with the influence of human factors in the construction of high-standard farmland projects, environmental pollution poses a serious threat to the rural ecological environment, and farmers are important actors in rural environmental protection.
Point 7: Suggestions – the name and position of this chapter, after Conclusions, is rather odd. It should be Discussions and this section should address in more detail the title of the paper – the role of environmental policy and farmers’ behaviour. It would be interesting to know if socio-demographic characteristics – age, gender and education influence environmental behaviour and to what extent.
Response 7: Thank you for your suggestion. We adopted the framework you recommended in the first draft, but the team members thought that we did not do multi-group analysis to study relevant issues, so we wrote it from a different Angle. Next, we removed the suggestion section and rewrote the discussion section.
Point 8: The last sentence of the abstract states that ‘This study could help to provide references for farmers to improve environmental protection behavior (line 28)’. I think it is rather preposterous that a scientific paper would reach directly farmers, when ¾ of the interviews farmers (that should be a representative sample) attended only primary or middle school according to the information presented.
Response 8: Thank you for your correction of the question. It is rather gilding the lily to add such a sentence in the abstract. We will delete the sentence.
Point 9: Line 13: ‘Based on 295 farmers in the Guanzhong Plain region,’ – I think it should be ‘based on interviews with 295 farmers’
- Line 19, 23: participation awareness – I think this needs rephrasing.
- 129: ‘they will be more intention to participate in it’ - rephrasing required
- 198-199: ‘As presented in Appendix A’- just (Appedix A).
- 223/ Table 1 – village cadres -what does it mean? There should be a short explanation.
- 247, 249: Cronbachis'a coefficients ‘’ – Cronbach’s α coefficient
- 363 : ‘ngos’ – NGOs
- 391-393 : ‘To actively respond to the opinions and suggestions put forward by farmers, to form a sound communication between the government and the people, and to implement the rural environmental protection funds, to actively participate in the farmers to give policy subsidies;’ There is no main sentence. Also, what does ‘actively participate in the farmers to give policy subsidies’ mean?!
Response 9: We have adjusted and modified this part in the revised manuscript.

Reviewer 2 Report
Please find the comment on the attachment

Author Response
Response to Reviewer 2 Comments
On behalf of my co-authors, we thank you very much for giving us an opportunity to revise our manuscript, we appreciate editor and reviewers very much for their positive and constructive entitled “The impact of farmer participation on environmental protection behavior in the era of rural Revitalization: The moderating role of policy environment”.
We have studied reviewer’s comments carefully and have made revision which marked in red in the paper. We have tried our best to revise our manuscript according to comments. Attached please find the revised version, which we would like to submit for your kind consideration.
We would like to express our great appreciation to you and reviewers for comments on our paper.
Looking forward to hearing from you.
Thank you and best regards.
Point 1: Provide theoretical justification for moderating effect (PC) in introduction section.
Response 1: As a recent reviewer, we have added the theoretical analysis of the moderating effect to the introduction of the revised manuscript.
Point 2: Revisit your literature and analyze it critically.
Response 2: We have added literatures to the revised manuscript and systematically analyzed their support for the theoretical hypothesis in the article.
Point 3: Since, hypothesis is testable relationship between variables, develop your hypotheses from association between variables not from a single construct.
Response 3: We rewrote the research hypothesis and elicited the correlation of the hypothesis from the theoretical demonstration. And systematically studied a number of PLS-SEM empirical analysis papers.
Point 4: A total of 400 questionnaires were distributed, how you collected 1295 as mentioned in section 3.4?
Response 4: Dear reviewer, 400 questionnaires were distributed in our study, and 295 valid questionnaires were collected. You should think of the 1 in Table 1 and 295 after it as 1295.
Point 5: Provide detail on how randomization was done, how sample size was determined?
Response 5: This study is based on the data of farmers in Baoji and Weinan, Guanzhong region, from April to September 2022. The reason why this research area is selected is that high-standard farmland projects are being built there, and the research area has accumulated rich experience in environmental protection. The villagers of the villages participating in environmental protection in the above areas were selected as the investigation objects. The survey was conducted from April to June and from July to September 2022. First, 30 farmers were selected from the field of high-standard farmland project for pre-survey, and then the validity of the questionnaire was tested and the snowball method was adopted to collect the questionnaire. The questionnaire was distributed by a combination of interviews and on-site questionnaires. The interviewees must be villagers or village officials involved in or familiar with environmental protection. In order to improve the authenticity of respondents filling in the questionnaire, and to clearly express the academic purpose and research value of the research to respondents during the research process, we promise respondents to fill in and handle the questionnaire anonymously, so as to ensure that the respondents are voluntary and free from relevant concerns, and ensure the reliability and persuasion of the results.
Point 6: I found from Table A1 that some of the relationship between construct and indicators is formative however, you have used it as reflective. I didn’t find any justification for this, provide valid justification for this under the rubric of measurement theory. For detail see (Petter et al., 2007).
Response 6: We have systematically studied the article "Specifying formative constructs in IS research using PLS and covariance-based techniques" recommended by you. Our indicators refer to mature questionnaire items in published articles and adopt reflective indicators. We measure questions for farmers and fill in questions 1 to 7 based on their feelings.
Point 7: Your research model as provided on page 3 is “with moderating effect”, however, you provided results of mediating model on page 9. Don’t you think omitting the explanatory power of moderating variable will affect the structural paths? For example on Table 6 page 9, the path value between SN and PB is .209 whereas the value for the same path in structure model with moderating effect is 0.166 (might be insignificant). The same is the case for other paths. Revisit and provide appropriate results based on your structural model with moderating effect.
Response 7: We are very sorry for the trouble caused to you due to the lack of rigor of our team. We have revised the data and model, and made detailed explanations and data explanations on the direct effect, mediating effect and moderating effect in the revised manuscript.
Reviewer 3 Report
Thank you for the opportunity to discover this interesting article. Below are my comments to improve the article.
1. Theoretical background. Fundamental articles on the concepts and their definitions used in the study are not cited, e.g. theory of planned behavior - Ajzen, I. (1991). The theory of planned behavior. Organizational behavior and human decision processes, 50(2), 179-211. I would suggest referring to the basic articles and citing the definitions of the studied constructs in the paragraph on the development of hypotheses.
2. Introduction is a sufficient introduction to the article. The hypotheses are correctly derived from the literature.
3. The research tools were well described and their reliability was tested.
4. The statistical part is very good. The results are correctly and accurately described.
5. The Discussion section is missing.
6. There is no description of the research limitations of the article.
7. Please consider expanding the bibliography.
Author Response
Response to Reviewer 3 Comments
On behalf of my co-authors, we thank you very much for giving us an opportunity to revise our manuscript, we appreciate editor and reviewers very much for their positive and constructive entitled “The impact of farmer participation on environmental protection behavior in the era of rural Revitalization: The moderating role of policy environment”.
We have studied reviewer’s comments carefully and have made revision which marked in red in the paper. We have tried our best to revise our manuscript according to comments. Attached please find the revised version, which we would like to submit for your kind consideration.
We would like to express our great appreciation to you and reviewers for comments on our paper.
Looking forward to hearing from you.
Thank you and best regards.
Point 1: Theoretical background. Fundamental articles on the concepts and their definitions used in the study are not cited, e.g. theory of planned behavior - Ajzen, I. (1991). The theory of planned behavior. Organizational behavior and human decision processes, 50(2), 179-211. I would suggest referring to the basic articles and citing the definitions of the studied constructs in the paragraph on the development of hypotheses.
Response 1: Dear reviewer, we have added basic literature and definitions to the theoretical background of the revised manuscript. In addition, we systematically reviewed the contributions of literature to the hypothesis of this study. In this study, based on the survey data of farmers in Guanzhong Plain under the framework of extended planned behavior theory, the partial least square structural equation model was used to explore the mechanism of the influence of psychological cognitive factors on farmers' PI and PB in environmental protection. It also reveals the moderating effect of PE on farmers' PI - PB.
Point 2: Introduction is a sufficient introduction to the article. The hypotheses are correctly derived from the literature.
Response 2: We have added literatures to the revised manuscript and systematically analyzed their support for the ideas in the article.
Point 3: The Discussion section is missing. There is no description of the research limitations of the article. Please consider expanding the bibliography.
Response 3: We have added a section on discussion and research limitations and future directions to the revised manuscript
Reviewer 4 Report
The manuscript is about the impact of farmer participation on environmental protection behavior in the era of rural revitalization: the moderating role of policy environment.
The reviewer has following comments:
1. Enrich the introduction section with more references to support the state of the art and justify your work.
2. Overall comments: the paper is well written and organized. I recommend for publication.
Author Response
Response to Reviewer 4 Comments
On behalf of my co-authors, we thank you very much for giving us an opportunity to revise our manuscript, we appreciate editor and reviewers very much for their positive and constructive entitled “The impact of farmer participation on environmental protection behavior in the era of rural Revitalization: The moderating role of policy environment”.
We have studied reviewer’s comments carefully and have made revision which marked in red in the paper. We have tried our best to revise our manuscript according to comments. Attached please find the revised version, which we would like to submit for your kind consideration.
We would like to express our great appreciation to you and reviewers for comments on our paper.
Looking forward to hearing from you.
Thank you and best regards.
Point 1: Enrich the introduction section with more references to support the state of the art and justify your work.
Response 1: We have added literatures to the revised manuscript and systematically analyzed their support for the ideas in the article.
Round 2
Reviewer 1 Report
Thank you for painstakingly replying to all comments and improving the manuscript. I think the current version can be published.